# Perfusion of MC3T3E1 Preosteoblast Spheroids within Polysaccharide-Based Hydrogel Scaffolds: An Experimental and Numerical Study at the Bioreactor Scale

**DOI:** 10.3390/bioengineering10070849

**Published:** 2023-07-18

**Authors:** Jérôme Grenier, Bertrand David, Clément Journé, Iwona Cicha, Didier Letourneur, Hervé Duval

**Affiliations:** 1Laboratoire de Génie des Procédés et Matériaux, CentraleSupélec, Université Paris-Saclay, 91190 Gif-sur-Yvette, Franceherve.duval@centralesupelec.fr (H.D.); 2Laboratoire de Mécanique Paris-Saclay, CNRS, CentraleSupélec, ENS Paris-Saclay, Université Paris-Saclay, 91190 Gif-sur-Yvette, France; 3Laboratoire de Recherche Vasculaire Translationnelle (LVTS), INSERM U 1148, Université Paris Cité, Université Sorbonne Paris Nord, Hôpital Bichat, 75018 Paris, France; clement.journe@inserm.fr (C.J.); didier.letourneur@inserm.fr (D.L.); 4Department of Otorhinolaryngology, Head and Neck Surgery, Section of Experimental Oncology und Nanomedicine (SEON), Friedrich-Alexander Universität Erlangen-Nürnberg, Universitätsklinikum Erlangen, 91054 Erlangen, Germany; iwona.cicha@uk-erlangen.de

**Keywords:** spheroids, organoids, perfusion bioreactor, porous hydrogel, 3D cell culture, lattice Boltzmann method, fluid dynamics, oxygen transport

## Abstract

The traditional 3D culture systems in vitro lack the biological and mechanical spatiotemporal stimuli characteristic to native tissue development. In our study, we combined porous polysaccharide-based hydrogel scaffolds with a bioreactor-type perfusion device that generates favorable mechanical stresses while enhancing nutrient transfers. MC3T3E1 mouse osteoblasts were seeded in the scaffolds and cultivated for 3 weeks under dynamic conditions at a perfusion rate of 10 mL min^−1^. The spatial distribution of the cells labeled with superparamagnetic iron oxide nanoparticles was visualized by MRI. Confocal microscopy was used to assess cell numbers, their distribution inside the scaffolds, cell viability, and proliferation. The oxygen diffusion coefficient in the hydrogel was measured experimentally. Numerical simulations of the flow and oxygen transport within the bioreactor were performed using a lattice Boltzmann method with a two-relaxation time scheme. Last, the influence of cell density and spheroid size on cell oxygenation was investigated. The cells spontaneously organized into spheroids with a diameter of 30–100 μm. Cell viability remained unchanged under dynamic conditions but decreased under static culture. The cell proliferation (Ki67 expression) in spheroids was not observed. The flow simulation showed that the local fluid velocity reached 27 mm s^−1^ at the height where the cross-sectional area of the flow was the smallest. The shear stress exerted by the fluid on the scaffolds may locally rise to 100 mPa, compared with the average value of 25 mPa. The oxygen diffusion coefficient in the hydrogel was $1.6\times 10^{-9}$ m^2^ s^−1^. The simulation of oxygen transport and consumption confirmed that the cells in spheroids did not suffer from hypoxia when the bioreactor was perfused at 10 mL min^−1^, and suggested the existence of optimal spheroid size and spacing for appropriate oxygenation. Collectively, these findings enabled us to define the optimal conditions inside the bioreactor for an efficient in vitro cell organization and survival in spheroids, which are paramount to future applications with organoids.

## 1. Introduction

For decades, in vitro cultures have been limited by flat, physiologically non-relevant environments. In vivo, the environment of cells is three-dimensional (3D). Cell–cell interactions are enhanced when cells organize themselves into 3D spherical clusters called spheroids, and later organoids [1]. A spheroid is a 3D assembly of cells, whereas an organoid exhibits a higher level of maturation with some physiologic parameters closer to the target organ. In this context, 3D cell structures have great potential for tissue engineering, which aims to replace damaged or diseased organs. Tissue engineering requires standardization of both the scaffolds (e.g., their porosity, chemistry, and controlled microarchitecture) and the cell culture conditions. The perception of the microenvironment is a crucial element in the self-organization of cells within spheroids/organoids [2]. Therefore, the in vitro reproduction of a physiological 3D microenvironment leading in vivo to the formation of tissue is both highly challenging and critically important for the physiological relevance of experimental models [3]. This mimicry is also a key factor in clinical translation. In addition to biological factors, spheroids/organoids formation and maturation are regulated by parameters such as fluid flow, mechanical stresses, and oxygen concentration [4,5]. Perfusion bioreactors provide a controlled environment for the in vitro study of the interactions between biomaterials and cells. They not only promote the exchange of nutrients and oxygen between the cells that colonize porous biomaterials and the culture medium, but also generate fluid shear stress known to induce cell proliferation and bone formation [6,7]. With regard to the 3D cell culture, some porous biomaterials for tissue engineering constitute particularly well-suited scaffolds [8,9]. For example, it has been previously shown that porous hydrogels promote the spontaneous formation of spheroids during seeding [10]. In this context, polysaccharide-based biopolymers represent good candidates for producing tissue engineering scaffolds. In addition, this type of hydrogel can easily be supplemented with bioactive molecules, such as hydroxyapatite, that induce in vivo bone tissue formation [11].

The chemical and mechanical environment experienced by the cells is not homogeneous within a bioreactor. Furthermore, without in situ local measurements, this environment is not known. Computational Fluid Dynamics (CFD) simulations are very useful for estimating the nutrient concentration fields and the stress field seen by the cells. Many studies are dealing with the simulation of stirred tank bioreactors since this type of bioreactor is most commonly used in the laboratory and industry. In these reactors, the cells are cultured in a suspended state or optionally on suspended microcarriers, and the flow regime is typically turbulent to enhance mass transfer. These reactors are well suited to producing active pharmaceutical ingredients, such as monoclonal antibody protein, by mammalian cells. The CFD simulations quantify the hydromechanical stress experienced by the cells [12], the mixing [13,14], and the oxygen transport and consumption [15,16]. Other modeling approaches focus on cellular metabolism; these models assume perfect mixing within the bioreactor and describe the microkinetic reaction network involved during the culture [17,18]. Perfusion bioreactors significantly differ from stirred tank bioreactors; they are actually fixed-bed reactors, and they are generally operated in a laminar regime. The bed consists of a single large porous scaffold or a stack of porous scaffolds. The cells grow within the pores of the scaffolds. Several studies focus on perfusion bioreactors for bone tissue engineering. The porous scaffolds are typically made of impermeable materials (as opposed to hydrogels). CFD simulations are used to assess the wall shear stress [19], the oxygen and glucose concentrations [20], and the cell growth rate [21,22,23] at the local scale within the reactor. 

The present study aims to evaluate an in vitro tissue engineering strategy based on a porous biomaterial within a perfusion bioreactor, which is characterized experimentally and numerically. For this purpose, we choose porous polysaccharide-based scaffolds that demonstrated robust improvement in bone formation during in vivo experiments [11,24]. We used MC3T3E1 mouse preosteoblasts, a mechanosensitive cell line that has been extensively studied in the context of osteoblastic differentiation [25,26].

Our study endeavors to compute the fluid dynamics and the corresponding shear stress inside the perfused bioreactor, and to correlate the results with the proliferative capacity of the cells forming the aggregates inside hydrogels enclosed within the bioreactor. Spheroids were identified thanks to pre-labeling cells with biocompatible superparamagnetic iron oxide nanoparticles coated with lauric acid and human serum albumin (SPION^LA-HSA^) [27]. The oxygen transport in the bioreactor was also considered in this study. Indeed, oxygen availability is of key importance for the success of bone grafting since hypoxia down-regulates bone differentiation [28]. Analytic resolution of the one-dimensional diffusion equation demonstrated that the characteristic diffusion length of oxygen limits the biomaterials’ thickness if a necrotic core is to be avoided [29]. This one-dimensional analysis, however, did not take into account cell organization inside the biomaterials [16]. In the present work, a parametric study was carried out by varying cell cluster size and cell number per scaffold to assess oxygen transport. The numerical simulations were compared to experimental viability measurement. Hydrodynamics and oxygen transport simulations were performed using lattice Boltzmann methods (LBM), which is a very attractive approach to address flows over complex boundaries such as scaffolds in perfused bioreactors [19,20,22]. The simulations were carried out based on the 3D acquisition of the hydrogel scaffolds inside the perfused bioreactor using MRI, which is a method of choice for tissue characterization [30]. 

Using the predictive nature of numerical simulations in terms of mass transfer species such as oxygen, this paper provides a rationale for the choice of initial cell culture conditions to ensure optimal and homogeneous cell viability within the whole spheroid-containing hydrogel. Current knowledge regarding, among others, the understanding of the influence of species transport and momentum mechanisms does not allow for a rapid and realistic translation of a bone products fabrication process to the clinic. The ambition of this paper was to deepen the knowledge and methodology in this field by carrying out various experiments and numerical simulations. More specifically, among our objectives was to ensure homogeneous perfusion to distribute oxygen, nutrients, and biological factors within the biomaterial and to provide spheroids with a defined, controlled mechanical environment capable of directing maturation on its own towards the osteoblastic pathway without altering its integrity and viability.

## 2. Materials and Methods

### 2.1. Porous Polysaccharide-Based Hydrogel Scaffolds Preparation

Porous scaffolds were synthesized according to Grenier et al. [31] from polysaccharide-based hydrogel made of pullulan (130 g L^−1^, Pullulan 200,000 Hayashibara) and dextran (43.3 g L^−1^, Dextran 500 kDa Pharmacosmos) that are crosslinked by sodium trimetaphosphate (STMP, 9.35 × 10^−2^ mol L^−1^) under saline conditions (NaCl, 3.46 mol L^−1^) and alkaline conditions (NaOH, 0.953 mol L^−1^). After crosslinking (20 min at 50 °C), hydrogel discs were formed (2.5 mm in radius and 1 mm thickness) by disc cutting. The samples were neutralized in phosphate buffer (DPBS 10×) and washed overnight in saline solution (NaCl, 0.250 g L^−1^) until equilibrium. The swollen hydrogel discs were placed (the base was horizontal) on Petri dishes (VWR, 391-0875) in batches of 50 before undergoing freeze-drying in a MUT freeze-drier (Cryotech^®^). Plates were initially kept at 15 °C, and the cooling rate was set to −0.1 °C min^−1^ until −20 °C. Freezing occurred after nucleation at –10 ± 2 °C [31]. The freeze-dried hydrogel disks (10 mm diameter, 1.8 mm thickness) were porous with a mean pore size of 230 ± 20 μm [31]. In the following, they are termed porous hydrogel scaffolds. X and Y (in capital letters) denote the direction parallel to the scaffold base, and Z denotes its symmetry axis. Figure 1 shows the porous structure of a scaffold in the dry state (Figure 1A) and in the swollen state (Figure 1B).

### 2.2. MC3T3E1 Cell Seeding in Porous Hydrogel Scaffolds

MC3T3E1 mouse osteoblasts (American Type Culture Collection, ATCC, Manassas, VA, USA) were cultivated in 2D monolayers and kept in the exponential growth stage on a culture plate (Techno Plastic Products AG, TPP, Trasadingen, Switzerland) under standard cell culture conditions (37 °C, 100% humidity, and 5% CO_2_). The culture medium contained α-MEM with nucleosides and without ascorbic acid (Gibco by Thermo Fisher Scientific France, Asnières-sur-Seine, France), supplemented by 10% (mg mL^−1^) bovine fetal serum (Dominique Dutscher, Dubernolsheim, France) decomplemented at 56 °C for 30 min and by 1% (mg mL^−1^) penicillin and streptomycin (10,000 μ mL^−1^–10 mg mL^−1^, PAN-Biotech GmbH, Aidenbach, Germany). After 15 min of incubation, the cells were detached with Trypsin/EDDTA 0.05/0.02% (PAN-Biotech GmbH, Aidenbach, Germany), washed with medium, and concentrated by centrifugation into suspensions of 5, 15, 20, and 30 million per mL. The porous hydrogel scaffolds, previously submitted to UV light (15 min on both sides, VL-215.G, Bioblock Scientific, Illkirch, France), were seeded with 20 μL of cell suspension (deposited on the top surface). Depending on the experiment, the final quantities of cells per scaffold were 100,000, 300,000, 400,000, or 600,000. Total swelling of the scaffolds was obtained by adding three times 20 μL of medium per scaffold. The size of the swollen scaffolds (8 mm in diameter, 1.4 mm in thickness) was smaller than in the dry state because the scaffolds were freeze-dried after swelling in NaCl (0.25 g L^−1^), and the swelling ratio of the hydrogel in this solvent is significantly higher than in DPBS [31]. 

For the 3D static culture, seeded porous hydrogel scaffolds were incubated under standard cell culture conditions in 6-well plates (4 scaffolds per well) with 1.5 mL of medium, refreshed every 3 days.

### 2.3. 3D Dynamic Culture in a Perfusion Bioreactor

Five hours or twenty-four hours post-seeding, 48 seeded porous hydrogel scaffolds were randomly stacked over a 3 cm height inside a polypropylene (PP) tube (12 mm inner diameter and 4 cm height) and held between two Teflon grids (5 mm height) with 100 holes of 1.2 mm in diameter and 50% porosity (Figure 2).

The macroporosity consists of the free space between the scaffolds. The PP tube was transferred to the bioreactor that consisted of a cylindric chamber made of polycarbonate (PC), which was perfused through flexible silicon tubes of 3.2 mm inner diameter (R6504—26BPT, PharMed, Saint-Gobain Tuyaux, Pont-à-Mousson, France) with 200 mL of culture medium equilibrated at pH = 7.2 by the bubbling of CO_2_-enriched air (5%) at a 15 mL min^−1^ air flow rate. A peristaltic pump (Easy-load 3, Masterflex, Cole-Parmer, Vernon Hills, IL, USA) allowed a perfusion rate or volumetric flow rate of q= 10 mL min^−1^. The so-called superficial velocity was then $v_{s}=$ 1.47 mm s^−1^ calculated through the empty PP tube cross-section. In the following, x and y (in small letters) denote the directions parallel to the PP tube transverse cross-section, and z denotes the direction parallel to the PP tube axis.

The PC cylindrical chamber was chemically sterilized with bleach (0.05 g L^−1^, 30 min), extensively washed with sterile water, and dried under a sterile atmosphere with an abundant quantity of 70% ethanol. The other elements of the bioreactor (PP tube, grids, and flexible tubes) were sterilized using the autoclave.

Culture started in static conditions (6-well plates, 4 hydrogels per well, 1.5 mL of culture medium). The dynamic conditions were run for at least 24 h post-seeding unless otherwise stated. In the experiments with longer time spans, the circulating culture medium was renewed every 7 days. Three perfused seeded scaffolds were sampled every one to three days during the three weeks of cultivation and placed temporarily in a 24-well plate (one scaffold per well) before being analyzed. For this purpose, the pump was stopped, the bioreactor was opened under a sterile atmosphere by removing the top grid, and scaffolds were sampled. Then, the bioreactor was closed again, and the perfusion was restarted after a less than 10 min interruption.

### 2.4. Evaluation of Cell Number, Division, and Viability by CLSM

#### 2.4.1. Cell Number per Scaffold

Confocal laser scanning microscopy (CLSM, LSM 700, Carl Zeiss, Rueil-Malmaison, France) was used to assess cell number alterations inside the porous hydrogel scaffolds. Seeded scaffolds were sampled, and cells were fixed with 4% paraformaldehyde (PFA, 250 μL per scaffold). Cell membranes were previously labeled with red dye PKH-26 (Sigma–Aldrich, St. Louis, MO, USA) according to the manufacturer's instructions. Hydrogel was pre-labeled by the addition of 0.17% (g L^−1^) 500 kDa FITC-dextran (TdB Consultancy). Seeded porous hydrogel scaffolds were analyzed in 3D over 370 μm depth by 26 XY slices with 2.5 μm resolution (XY is parallel to the scaffold base). Images analysis was performed in 3D with Fiji (ImageJ 1.54d, National Institutes of Health, Bethesda, MD, USA) after thresholding to extract cell volume with the 3D Object Counter plugin. The cell number over a 370 μm height was estimated from the volume of a single MC3T3E1 cell (3000 μm^3^, [32]). This number was extrapolated to the total cell number inside the whole scaffold (1.4 mm in height). The cumulative distribution of spheroid diameters ($d_{s}$) was fitted by the cumulative distribution function of the exponential law:(1)$$F\left(d_{s}\right)=1-e^{-(d_{s}-d_{min})/\delta }$$
where $d_{min}$ was systematically set to the equivalent diameter of a single cell, i.e., 18 μm and δ (μm) were adjusted.

#### 2.4.2. Cell Proliferation

The expression of cell proliferation antigen Ki67 was assessed by immunofluorescence. For this purpose, the cells within the porous hydrogel scaffolds were fixed with 4% PFA (250 μL per scaffold) and permeabilized with 0.1% Triton X (three times 30 min incubation of 250 μL per scaffold). Cells were incubated at 4 °C overnight with 250 μL (8 μg mL^−1^) of primary polyclonal rabbit anti-Ki67 antibodies per scaffold (ab15580). After washing with 250 μL of DPBS (three times), cells were incubated at 25 °C for 30 min with 250 μL of secondary donkey anti-rabbit antibodies conjugated with Alexa Fluor^®^ 647 nm (ab150075, 4 μg mL^−1^). Cell nuclei were stained with DAPI (ThermoFisher, Waltham, MA, USA, 250 μL at 1 μg mL^−1^) for 30 min at 25 °C. Cells inside the porous scaffolds were analyzed by CLSM with 1.25 μm resolution.

#### 2.4.3. Cell Viability Assessment

Live/Dead kit (Invitrogen by Thermo Fisher Scientific France, Asnières-sur-Seine, France) was used to evaluate the viability and mortality of the cells. Seeded hydrogel scaffolds were incubated for 30 min under standard cell culture conditions with 250 μL of culture medium that contained calcein-AM (2 μmol L^−1^) and ethidium homodimer-1 (5 μmol L^−1^). After washing with 250 μL of DPBS (three times), cells were fixed with 4% PFA (250 μL per scaffold). Inside the porous scaffolds, representative regions were analyzed by CLSM with 1.25 μm resolution, and the whole scaffolds were analyzed over 370 μm depth with 26 XY slices at 2.5 μm resolution.

### 2.5. MRI Acquisitions

#### 2.5.1. Cell Labelling with Superparamagnetic Iron Oxide Nanoparticles

To enhance MRI contrast, the cells were labeled for 36 h with biocompatible superparamagnetic iron oxide nanoparticles (SPION^LA-has^) produced according to Zaloga et al. [27] (see Appendix A for details). After washing, the cells were seeded on top of the porous hydrogel scaffolds, as described in Section 2.2. Seeded porous hydrogel scaffolds contained 100,000, 300,000, or 600,000 cells per scaffold. Some seeded scaffolds were immediately placed in a 2 mL centrifuge tube (Eppendorf France, Montesson, France), and cells were fixed with 4% PFA. A total of 48 scaffolds seeded with 1000,000 cells per scaffold were stacked in the perfused bioreactor and cultivated under 3D dynamic culture conditions, as described in Section 2.3, at a perfusion rate of q= 10 mL min^−1^.

#### 2.5.2. MRI Sequences

A 7 Tesla MRI (Pharmascan 70/16, Bruker, Billerica, MA, USA) equipped with ParaVision v6.0.1 software (Bruker BioSpin GmbH, Rheinstetten, Germany) was used in this study (see Appendix A for details). 

“T2-turbo RARE” sequences were used to obtain bioreactor images. For one bioreactor, two acquisitions were performed. Each acquisition consisted of 135 xy transverse slices (we remind that z-direction is parallel to the bioreactor axis) of 250 μm thickness and 55 μm^2^ resolution. The second acquisition was a duplicate of the first one after a translation of 125 μm in the z-direction. The two acquisitions lasted 70 min. A 125 μm resolution in z-direction was then achieved by merging the two acquisitions. After a polynomial interpolation in z-direction, a 3D image with 55 μm resolution in the three directions was obtained.

“$T_{2}^{*}$-MSME (Multi-Slice-Multi-Echo)” sequences were used to quantify (by relaxometry) the concentration of SPION^LA-HSA^. Sequences were performed on 250 μm thick slices with a 55 μm resolution. A total of 11 echo times (T_E_) with 13,845 ms spacing were used. The $T_{2}^{*}$ coefficients were estimated by fitting with the exponential function proposed by Milford et al. [33] (see Appendix A for details).

### 2.6. Measurement of the Oxygen Diffusion Coefficient inside the Hydrogel

#### 2.6.1. Experimental Setup

A dedicated experimental setup inspired by Hulst et al. [34] was developed to measure the oxygen diffusion coefficient in the polysaccharide-based hydrogel (Figure 3).

The centerpiece of the device was a two-compartment reactor. The main chamber was a cylinder with an inner diameter of 36 mm and a total working volume of $V_{r}=$ 84 mL, well-stirred with a magnetic barrel at 400 rpm. This chamber was connected through a 3D-printed grid (Eastman Amphora 3D Polymer AM3300, Eastman Chemical Company, Kingsport, TN, USA) of 2 mm thickness to a second smaller chamber of inner diameter d= 30 mm and height e= 4.5 mm. The sealing of the reactor was ensured by clamping systems with flat seals and fixing screws. Liquid circulated in the main chamber thanks to a peristaltic pump (AxFlow, Dublin, Ireland) and flexible hoses connected with 6 mm automatic push-in fittings (Parker Legris Hannifin Manufacturing France, Rennes, France). Liquid entered the bottom of the main chamber via a ball valve and exited at its top. Experiments were typically performed with DPBS supplemented with 1% (mg mL^−1^) penicillin–streptomycin antibiotics. A total of 500 mL DPBS supply was well-stirred and in equilibrium with air. The whole setup was kept at 37 °C using a water bath. 

An optical fluorescent oxygen sensor (OptiOx, inLab Mettler-Toledo SAS, Viroflay, France) equipped with an acquisition unit (Seven2Go, Mettler-Toledo SAS, Viroflay, France) and Easy Direct Ph software (Mettler-Toledo SAS, Viroflay, France) was used to acquire every 6 s the dissolved oxygen concentration of the liquid phase in the main chamber of the bioreactor. The second chamber was filled with the crosslinked polysaccharide-based hydrogel to be characterized. The hydrogel volume was $V_{h}=\pi e{\left(d/2\right)}^{2}$.

#### 2.6.2. Measurement Protocol

Dissolved oxygen was removed from the liquid phase of the reactor by nitrogen bubbling for 10 min until the oxygen concentration measured in the main chamber had decreased below 0.2 mg L^−1^. Then, trapped nitrogen bubbles were chased away by adding DPBS from the 500 mL supply, and oxygen concentration increased typically around 1 mg L^−1^_._ The reactor was closed for 40 min until the oxygen concentration of the liquid phase reached the plateau value denoted $c_{1}$. A stepwise increase in oxygen concentration was imposed by rapid renewal (30 s) of the reactor liquid medium thanks to the pumping of 200 mL of oxygen-saturated DPBS. The oxygen concentration measured in the main chamber reached the maximal value denoted $c_{2}$. The time origin was set at the beginning of the step. The reactor was again isolated from the circuit, and the decreasing oxygen concentration c(t) was recorded inside the main chamber until its stabilization at the plateau value denoted $c_{\infty }$.

#### 2.6.3. Identification of the Oxygen Diffusion Coefficient

Experimental oxygen concentration data were analyzed with the unsteady diffusion model proposed by Carman and Haul [35]. The hydrogel was considered a solid medium of thickness e exchanging oxygen through one of its faces with the well-stirred liquid volume inside the main chamber. Zero-flux boundary condition was assumed on the other faces. Under these assumptions, the dissolved oxygen concentration c(t) was expected to follow the simplified equation:(2)$$\ln\left(\frac{c\left(t\right)-c_{\infty }}{c_{2}-c_{\infty }}\right)≅\ln\left(A\right)-\frac{BD}{e^{2}}t$$
where D is the oxygen diffusion coefficient in the hydrogel, and A, B are constants that depend only on $c_{1},\;c_{2}$, and $c_{\infty }$ (see Appendix A for more details). Thus, the oxygen diffusion coefficient can be determined by linear regression (after appropriate formatting of the experimental data).

Furthermore, at equilibrium, the mass conservation given for the dissolved oxygen is:(3)$$V_{r}\left(c_{2}-c_{\infty }\right)=V_{h}\left(Kc_{\infty }-K\;c_{1}\right)$$
where K is the oxygen partition coefficient between the hydrogel and the DPBS.

### 2.7. Computational Fluid Dynamic (CFD) Simulation of the Perfusion Bioreactor

#### 2.7.1. Modelling and Lattice Boltzmann Method Implementation

The culture medium was assumed to be incompressible, with fluid density ρ=993 kg m^−3^, and Newtonian, with constant dynamic viscosity $\mu =10^{-3}\;$ Pa s at 37 °C [36]. In this case, the fluid flow is described by the continuity equation and the Navier–Stokes momentum equations. The walls of the bioreactor, the hydrogel scaffolds, and the cells were considered impermeable solids, i.e., the fluid velocity equaled zero at the fluid/solid interface. Fluid flow was negligible inside the porous hydrogel scaffolds, as compared to the flow through the macropores of the scaffold stack. A two-relaxation time Lattice Boltzmann Method (TRT-LBM) with a cubic lattice in three dimensions and nineteen velocities (D3Q19) was used to simulate the Navier–Stokes equations [37]. Detailed implementation of the method was described by Duval et al. [38] and by Thibeaux et al. [39]. In the following, we summarize the general aspects of the method and introduce the underlying parameters. 

In this LBM, the fluid is described by a population of fictitious particles with a discrete and finite velocity distribution of Q=19 vectors. The particles are constrained to move along the 3D lattice. At each time increment, the particle population follows a two-step dynamic: First, the particles propagate according to their velocity from one lattice node to the other. Second, the particle velocity distribution relaxes toward the equilibrium distribution. The fluid velocity is given by the first-order moment of the particle velocity distribution. 

The relaxation involved two parameters. First, $s_{1}$ is directly related to the kinematic viscosity of the real fluid ν=μ/ρ by:(4)$$s_{1}={\left(\frac{\nu }{c_{s}^{2}}\frac{∆t}{{∆x}^{2}}+\frac{1}{2}\right)}^{-1}$$
where $c_{s}$ is the intrinsic speed of sound of the LBM ($c_{s}^{2}=1/3$ in lattice units), ∆x is the lattice spacing (in real units), and ∆t is the time step (in real units). Second, $s_{2}$ is computed from $s_{1}$ with the following so-called “magic number relation” [40]:(5)$$s_{2}=8\frac{2-s_{1}}{8-s_{1}}$$

The no-slip condition at the solid boundaries was modeled using bounce-back reflection. Since the two relaxation parameters satisfy Equation (5), the solid walls are located halfway between a fluid node and a solid node [40]. Finally, the time step was selected to keep the effects of LBM intrinsic compressibility negligible:(6)$$∆t=0.06∆x/v_{max}$$
where $v_{max}$ is the maximal fluid velocity.

To determine the wall shear stress that the fluid exerts, the viscous stress tensor $\;\overset{̿}{\sigma }$ and the wall normal vector $\vec{n}$ are required. The modulus of the wall shear stress τ is indeed given by:(7)$$\tau =\left|\overset{̿}{\sigma }\cdot \vec{n}-\left(\left(\overset{̿}{\sigma }\cdot \vec{n}\right)\cdot \vec{n}\right)\vec{n}\right|$$

The viscous stress tensor was calculated at each neighbor node of a solid surface (hydrogels or bioreactor lateral inner walls) from the LBM velocity distribution function at that node [39,41]. The determination of the boundary normal vector was not as straightforward, due to the staircase shape of the solid boundary. We implemented a technique proposed by Stahl et al. [41] to detect the normal wall direction locally. This method relies on the fluid flow properties near the solid boundary [41].

#### 2.7.2. Simulation of a Segment of the Perfusion Bioreactor

A segment of 6.55 cm in height was extracted from the reconstructed volume of the perfusion bioreactor. The CFD was performed on a 3D geometry that consisted of $n_{x}\times n_{y}\times n_{z}=220\times 220\times 149$ voxels with 55-μm resolution. The superficial velocity at the bioreactor inlet was set to $v_{s}=$ 1.47 mm s^−1^. The time step ∆t was computed by Equation (6), where $v_{max}=40\;v_{s}$. A numerical simulation consisted of two steps, i.e., a first run without the inertial terms (viscous regime) over 100,000 iterations, and a second one where the inertial terms were taken into account (simulation of the full Navier–Stokes equation). The first run provided a suitable initialization of the second one. The second run required about 120,000 other iterations until convergence. Results were exported and visualized using Tecplot (Tecplot 360 EX 2016).

### 2.8. Simulation of the Oxygen Transport in the Hydrogel

#### 2.8.1. Modelling and LBM Implementation

Oxygen transport within the perfusion bioreactor was described by the convection–diffusion equation. Because dissolved oxygen was highly diluted in the fluid, the diffusion flux was assumed to follow Fick’s law. The source term that corresponds to the consumption of oxygen by the cells per unit of volume and time was modeled according to a Michaelis–Menten-like kinetics:(8)$$\varphi =\frac{V_{max}}{\vartheta }\frac{c}{c+K_{M}}$$
where c (mol m^−3^) is the local dissolved oxygen concentration, $V_{max}$ (mol cell^−1^ s^−1^) is the maximal oxygen consumption rate, $K_{M}$ (mol m^−3^) is the Michaelis–Menten-like constant, that is to say, the oxygen concentration where oxygen consumption is reduced by 50%, and ϑ= 3000 μm^3^ is the volume of one MC3T3E1 cell [32]. Without relevant data for MC3T3E1 preosteoblasts, the following constants were estimated from NIH3T3 fibroblasts: $V_{max}=4\times 10^{-17}$ mol cell^−1^ s^−1^ [42] and $K_{M}=6\times 10^{-3}$ mol m^−3^ [43].

In the fluid, the dissolved oxygen diffusion coefficient was $D_{0}=3.0\times 10^{-9}$ m^2^ s^−1^, assuming pure water at 37 °C as a medium. In the hydrogel, the oxygen diffusion coefficient D was measured according to the method described in Section 2.6.2. In the cells, the oxygen diffusion coefficient was assumed to be D. The oxygen partition coefficients between the fluid and the hydrogel, and between the hydrogel and the cells, were assumed to equal 1.

A TRT-LBM with a cubic lattice in three dimensions and 7 velocities (D3Q7) were used to simulate the convection–diffusion equation as proposed by Ginzburg [44]. The dissolved oxygen was represented by a population of fictitious particles, distinct from that representing the fluid. The oxygen concentration is given by the zeroth order moment of the particle velocity distribution. The implementation of the TRT-LBM is detailed in the Appendix A. The main parameters of a simulation are summed up hereafter.

The relaxation of the particle velocity distribution depended on two parameters, $s^{-}$ and $s^{+}$. $s^{-}$ is related to the oxygen diffusion coefficient (D in the hydrogel, $D_{0}$ in the fluid). In the hydrogel, $s^{-}$ is given by
(9)$$s^{-}={\left(\frac{D∆t}{c_{e}\;{∆x}^{2}\;}+\frac{1}{2}\right)}^{-1}$$
where $c_{e}$ is a scale parameter fixed at $c_{e}=1/4$. The relaxation parameter $s^{+}$ satisfies the “magic number relation,” which presently expresses as $s^{+}=2-s^{-}$ [44]. The time step was chosen to ensure the stability of the D3Q7 LB method [44], which implies:(10)$$∆t\leq \frac{∆x}{2v_{max}}\;$$

Simulations were run until the oxygen concentration field reached a steady state. Results were exported and visualized using Tecplot 360 EX 2016 (Genias Graphics GmbH & Co., Regenstauf, Germany).

#### 2.8.2. Case Studies

The hydrogel scaffolds were numerically seeded with spheroids of a diameter of $d_{s}=80$ μm, 135 μm, or 400 μm. The total number of cells was adjusted to 100,000, 400,000, or 900,000 cells per scaffold. We introduced $c_{0}$, the concentration of dissolved oxygen when the culture medium was in equilibrium with air at 1 atm. We assumed that $c_{0}=0.21$ mol m^−3^, corresponding to the oxygen equilibrium concentration in pure water at 37 °C. Oxygen transport was studied in the two following configurations:Oxygen transport at the scaffold scale.

The simulation domain was reduced to a parallelepipedal scaffold of 4 mm width in the X-direction. At X=0 mm, the oxygen concentration was set at $c=c_{0}$. At X=4 mm, oxygen flux was set to zero (symmetry plane at X=4 mm). Periodic conditions were set in Y and Z directions mimicking a regular stack of scaffolds. The lattice spacing was ∆X = 10 μm. Spheroids of the same size were evenly spaced in the hydrogel scaffolds. The spacing between the spheroids was adjusted to meet the required number of cells per scaffold. The influence of the spheroid size and the number of cells per scaffold on the oxygen concentration profile was studied. 

bOxygen transport at the perfusion bioreactor scale.

The same section of bioreactor used for CFD (Section 2.7.2) was numerically seeded with spheroids of 135 μm diameter and 400,000 cells per scaffold. The lattice spacing was ∆x = 55 μm. The spheroids were randomly placed within the scaffolds according to a uniform distribution. The oxygen concentration in the fluid was assumed to be homogenous and equal to $c_{0}$ (we will return to this assumption in the discussion). Oxygen transport was simulated in the stacked hydrogel scaffolds.

### 2.9. Data Analysis

For all experiments, the values are given as mean (points) ± standard deviation of the mean calculated for at least 3 scaffolds from the same culture conditions (static or bioreactor). It happens that the error bars are smaller than the point size. The viability was computed for at least 200 spheroids per scaffold (n = 1).

## 3. Results

### 3.1. Dynamic vs. Static 3D Cell Culture Conditions

#### 3.1.1. Assessment of Cell Number

The number of cells per porous hydrogel scaffold was evaluated up to 22 days of culture under static or dynamic conditions at a perfusion flow rate of 10 mL min^−1^ (Figure 4A). Each freeze-dried scaffold was seeded with 100,000, 300,000, or 600,000 cells. Twenty-four hours post-seeding, the scaffolds contained predominantly spheroids, and isolated (non-clustered) cells represented a negligible part of the total cell number. The mean diameter of spheroids was 35 μm, 55 μm, and 60 μm, depending on initial seeding density and corresponding to an estimate of 8, 29, and 37 cells per cluster, respectively. 

The cell numbers were calculated using stack confocal microscopy images of PKH-26 pre-labeled cells. The numbers counted on 26 sections of 2.5 μm and extrapolated to the total thickness of the swollen scaffold (1.4 mm) were 40,000 ± 9000, 100,000 ± 65,000, and 210,000 ± 50,000 cells, respectively.

The equivalent diameter of the spheroids ranged from 18 μm (isolated cell) to more than 200 μm. The cumulative distribution (number-average) of spheroid diameters 24 h post-seeding determined for an initial seeding of 600,000 cells per scaffold is shown in Figure 4B. The first quartile, the median diameter, and the third quartile were equal to 26 μm, 38 μm, and 57 μm, respectively. The data are well described by Equation (1) with δ=28 μm. In order to evaluate the fraction of isolated cells, we also computed the volume-average distribution of spheroid diameter (Appendix A): we found that the fraction of isolated cells was lower than 1% of the total cell number and 50% of the cells belonged to spheroids greater than 85 μm in diameter.

In static conditions, the estimated number of cells decreased to 1000 cells per scaffold after 21 days of culture when the initial number of seeded cells was 100,000, while the estimated number of cells was 10,000 ± 3000 cells after 22 days of culture under dynamic conditions. For an initial seeding of 300,000 and 600,000 cells, the number of cells cultured under dynamic conditions remained constant for 19 days (100,000 ± 32,000 and 230,000 ± 64,000 cells, respectively). The cumulative distributions (number–average) of spheroid diameters looked similar over time. However, a slight coarsening was noted. For example, for an initial seeding of 600,000 cells per scaffold, the (number-average) median spheroid diameter increased from 38 μm to 46 μm after 14 days of dynamic culture (Figure 4B), and 50% of cells belonged to spheroids greater than 110 μm in diameter (Appendix A).

#### 3.1.2. Assessment of Cell Proliferation

Ki67 expression of cells was assessed by immunofluorescence (Figure 5). The cells in the exponential growth phase, freshly detached, seeded, and immediately analyzed, expressed the Ki67 protein (Figure 5(A2)). We observed that the cells formed spheroids in the pores of the porous scaffold within 5 h and still expressed the Ki67 protein at that time (Figure 5(B2)). After 24 h (Figure 5(C2,D2)) and 7 days (Figure 5(E2,F2)) of culture under static or dynamic conditions with an initial seeding of 400,000 cells per scaffold (dynamic conditions run 5 h post-seeding), the cells no longer expressed the Ki67 protein, in agreement with the above results of estimated cell numbers that remained constant for 19 days in dynamic conditions.

#### 3.1.3. Assessment of Cell Viability

Cell viability within porous hydrogel scaffolds seeded with 400,000 cells per scaffold was assessed by confocal microscopy using a Live&Dead assay (Figure 6). After 24 h of culture under static (Figure 6(A1–A3)) or dynamic (Figure 6(B1–B3)) conditions (dynamic conditions initiated 5 h post-seeding), the cell viability was 80% (representing less than 10 dead cells per cluster, average diameter of 70 μm corresponding to about 60 cells). After 7 days of culture under static conditions (Figure 6(C1–C3)), the viability decreased to 50%, whereby the dead cells were homogeneously distributed in the clusters. In contrast, dead cells were less numerous after 7 days of culture under dynamic conditions (about 10 per spheroid) and were mostly concentrated in the center of the clusters, surrounded by viable cells (Figure 6(D1–D3)). In dynamic cell culture conditions, the viability was 89%. The viability of the cells in spheroids did not depend on their position within the hydrogel scaffolds.

### 3.2. Assessment of Cell Number by MRI within the Porous Hydrogel Scaffolds

The porous hydrogel scaffolds seeded with SPION^LA-HSA^-labeled cells were analyzed by MRI. With the T_2_^*^-MSME relaxometry sequences, no correlation between T_2_^*^ and the seeded cell number was observed (see Appendix A for details). We inferred that the echo times T_E_ were not small enough to capture the relaxation signal generated by the cells [45]. Nevertheless, the total volumes of the cells could be estimated using a T2-turbo-RARE sequence and image processing (Figure 7). A calibration curve was successfully obtained by linear regression (R^2^ = 0.99) with the seeded cell number (Figure 7D). A bioreactor containing 48 cellularized scaffolds cultured for 18 days at a perfusion flow rate q=10 mL min^−1^ was then analyzed by MRI. With an initial cell number of 100,000 per scaffold, the number of cells per scaffold estimated by MRI using the calibration curve was 88,000 (Figure 7D), which again indicated the conservation of cells within the scaffolds cultured in dynamic conditions.

### 3.3. Oxygen Diffusion Coefficient within the Hydrogel

The oxygen concentration c (mg L^−1^) of the liquid phase in the well-stirred reactor was measured by an oxygen sensor as a function of time. The oxygen diffusion coefficient within the hydrogel, D, was computed using Equation (2) from the oxygen concentration decrease within the measurement chamber (Figure 8) using the slope of the linear regression (R^2^ = 0.985) of $-\ln\left(c(t)-c_{\infty }\right)$ against time. The underlying hypotheses were satisfied (see Appendix A for details). The whole experiment was repeated twice, showing that $D=\left(1.6\pm 0.5\right)\times 10^{-9}$ m^2^ s^−1^ with a partition coefficient K= 9 ± 1.

### 3.4. Digitalization of the Bioreactors by MRI Acquisitions

The grayscale MRI images (Figure 9(A1), resolution = 55 μm) of the 48 porous hydrogel scaffolds stack were segmented by image processing into the fluid phase and the solid phase (hydrogel scaffold) (Figure 9(A2)). The pores of the scaffolds (poorly interconnected when the scaffolds are in the swollen state) were neglected and assimilated to the solid phase. The digital hydrogel scaffolds were virtually seeded with 135-μm diameter spheroids represented by eight 55-μm voxels (Figure 9(A3)).

### 3.5. Numerical Simulations of Hydrodynamics and Oxygen Transport Using LB Methods

#### 3.5.1. Perfusion Flow through the Stack Acquired by MRI

The perfusion flow was simulated in 3D (Figure 10) within a section of the bioreactor that was acquired by MRI (Section 2.5.2) and showed 24 ± 2% porosity. The superficial fluid velocity was 1.47 mm s^−1^. The highest velocities were in the z-direction parallel to the bioreactor axis (Figure 10(A1))**,** and the maximal interstitial fluid velocity was 27.29 mm s^−1,^ where the local restriction in the flow area was located. The *z*-components of the velocities were highly heterogenous, and prevailing paths were clearly visible (Figure 10(A1)). In the *xy* transverse plane, the components of the fluid velocities were much lower, and they did not show any preferred direction Figure 10(A2). The wall shear stress at the interface between the hydrogel and the fluid or between the bioreactor inner walls and the fluid ranged from 0.1 mPa to 100 mPa with 25 mPa on average.

#### 3.5.2. Effect of Spheroid Size and Number on Oxygen Level in the Scaffold

Oxygen transport in the hydrogel was simulated in 3D for model configurations with evenly spaced identical spheroids. Configurations differed according to the spheroid size and the spacing between spheroids (Figure 11). The spacing was adjusted to meet a targeted number of cells per scaffold. In a steady state, the oxygen concentration in the spheroid cores decreased along the *X*-axis, and the spheroids close to the surface had higher access to the oxygen. At each depth, the simulated oxygen concentration was higher in the hydrogel seeded with 100,000 cells per scaffold (Figure 11(B1,B2,D1,D2)) than in the hydrogel seeded with 900,000 cells per scaffold (Figure 11(A1,A2,C1,C2)).

Hypoxia was assumed for $c<K_{M}=6\times 10^{-3}$ mol m^−3^, that is to say, when the activity of the cells was lower than $V_{max}/2$. For the hydrogels seeded with 900,000 cells per scaffold distributed as 80-μm spheroids spaced 200 μm apart, spheroids located deeper than X=1.2 mm were entirely hypoxic (Figure 11(A1,A2)). For the same number of cells per scaffold arranged in 400 μm, spheroids spaced 1 mm apart, hypoxia occurred beyond X=2 mm (Figure 11(C1,C2)). For the hydrogels seeded with 100,000 cells per scaffold in the form of 80 μm, spheroids spaced 400 μm apart; spheroids did not experience hypoxia (Figure 11(B1,B2)). However, for the same number of cells arranged in 400 μm, spheroids spaced 2 mm apart, the spheroid located deeper than X=2 mm was partially hypoxic, i.e., the core suffered hypoxia, and the shell did not (Figure 11(D1,D2)).

#### 3.5.3. Oxygen Transport at the Perfusion Bioreactor Scale

Oxygen transport was simulated in a segment of the perfusion bioreactor (Figure 10). Oxygen concentration in the fluid was set to 0.21 mol m^−3^. Spheroids of diameter $d_{s}=135$ μm were randomly placed within the scaffolds according to a uniform distribution. The number of spheroids was adjusted to meet a cell number of 400,000 per scaffold.

The oxygen concentration was greater than $K_{M}$ in more than 98% of the scaffold volume (hydrogel and spheroids). Only 2% of the cells suffered from hypoxia. The hypoxic region was located deep inside the stacked hydrogel scaffolds, and minimal oxygen concentration reached 0.004 mol m^−3^.

## 4. Discussion

### 4.1. 3D Dynamic Cell Culture Conditions

The self-organization of cells in 3D results in the formation of spheroids. This process leads to the formation of 3D structures of different morphologies, biological functionalities, and degrees of complexity. We have chosen to use a hydrogel with controlled porosity that promotes the spontaneous formation of spheroids during their initial seeding with cells. Most biomaterials promote cell adhesion and spreading [46,47]. Conversely, MC3T3E1 preosteoblasts seeded within our porous hydrogel scaffolds spontaneously form spheroids in less than 5 h (Figure 5B and Figure 6B). This behavior is similar to that observed by Napolitano et al. [48] in fibroblasts seeded in 6-well plates coated with polysaccharide hydrogel (non-adhesive agarose). Under those conditions, fibroblasts formed spheroids in diameters ranging from 50 to 100 μm, depending on the initial numbers of seeded cells, as also observed in our study. The initial culture conditions are known to determine the chances of a group of isolated cells forming functional organoids [49,50]. In particular, the initial size and shape of the aggregates are of paramount importance.

Spheroids/organoids can be maintained in culture at good viability only for a limited period of time. It is reported, for example, that epithelial organoids have a lifespan of about one week, which is insufficient to form a mature, fully functional organoid. In vivo, organogenesis processes take several weeks or even months for specific tissues or organs. In vitro, access to nutrients is critical for maintaining organoids in culture over a long period. The nutrients must diffuse to the center of the organoid, while the waste products must be able to leave it. Thus, it is reported that for brain organoids reaching several millimeters, the insufficient diffusion of nutrients to the center of the organoid causes a significant decrease in cell viability in its core [51,52]. In vitro, it is necessary to implement advanced strategies to maintain these organoids in culture for such periods as to allow their maturation [53,54]. To address this, we cultured preosteoblast spheroids in 3D within the porous hydrogel scaffold under static and dynamic conditions. Cell seeding on the freeze-dried scaffolds results in spontaneous cell infiltration, migration, and clustering through the pores of the swollen hydrogel, as described by Grenier et al. [55]. Interestingly, the cells were almost exclusively in the form of spheroids, and the fraction of isolated cells was negligible (less than 1%). The estimated number of cells 24 h post seeding was thus found to be lower than the total number of seeded cells. Even if the majority of the cells is expected to remain trapped in the pores [55], part of the cells could migrate outside the scaffold before forming the spheroids. The precise cell numbers in 3D are difficult to obtain, even if the scaffolds are transparent, as in our case. Although the calculation of cell numbers from stacks of confocal images of scaffolds with thicknesses greater than 1 mm could be insufficiently accurate for obtaining the exact cell numbers, they at least enable us to follow the trends with time and the comparison between static and dynamic conditions. We observed that the higher the initial cell density, the higher the number of cells retained. Since the present scaffolds are 3D-structured instead of 2D-flat, more than 1 million cells could be loaded on each small scaffold (10 mm diameter, 1.8 mm thickness). Importantly, the 3D dynamic cell culture conditions (perfusion flow rate 10 mL min^−1^) allowed to preserve the cell viability above 80% in the spheroids up to 7 days of culture (Figure 6), while the numbers of dead cells increased under static cell culture conditions (Figure 4 and Figure 6). This result is consistent with the ability of perfusion bioreactors to improve nutrients and oxygen transfers [56,57].

Mechanical parameters such as shear stresses have an impact on morphogenesis and tissue homeostasis [4,5]. For example, the shear stresses induced by blood flow on the endothelium trigger the functions of endothelial cells [58]. Likewise, the perfusion of renal organoids not only improves their maturation, but also promotes the formation of a vascular network [59]. Such biomechanical stimuli at the tissue level are lacking in traditional 2D cultures. Our approach consists of combining the use of porous 3D hydrogels with a bioreactor-type perfusion device that generates favorable mechanical stresses while controlling nutrient transfers. In bone tissue engineering, perfusion bioreactors are used to promote cell proliferation or differentiation thanks to shear stress [60]. Dynamic culture conditions for osteoblasts in 3D porous scaffolds or in bioreactors were reported to enhance the proliferation and differentiation of osteoblasts [61,62,63]. However, no cell proliferation was observed in our study for up to 21 days in dynamic cell culture conditions (Figure 4). The cells were still alive but did not proliferate, and osteoblasts cultured in the perfusion bioreactor showed no expression of Ki67 (from day 1 to day 7) (Figure 5). Interestingly, cell seeding at numbers greater than 100,000 (i.e., 300,000 to 600,000 cells) per porous hydrogel scaffold allowed to keep the number of cells, as well as the size of spheroids constant for up to 18 days of culture under dynamic conditions (Figure 4 and Figure 9B). We thus hypothesize that cells in spheroids divided enough to counterbalance cell death [64]. By seeding a higher number of MC3T3E1 preosteoblasts, larger spheroids with highly viable cells were obtained. Ivanov et al. [65] also report a « bell-shaped » relationship between the proliferation and spheroids’ size of tumoral neural cells: up to a limit in size, cell proliferation increases with the size of the spheroids.

The lack of cell proliferation is mostly attributed to the lack of cell adhesion on the hydrogels. Pullulan and dextran polysaccharides that constitute the scaffolds are highly hydrophilic. This property does not allow protein adsorption [66,67]. However, proliferation in pullulan/dextran hydrogels has been reported in studies with other cell types, such as endothelial cells [68,69], megakaryocytes [10], and immortalized spheroids [8]. This suggests that proliferation ability within this hydrogel also depends on the cell type and/or scaffold properties. Another hypothesis is that the fluid flow through the pores of the hydrogel scaffolds is too weak to produce significant shear stress on the entrapped spheroids of MC3T3E1 preosteoblasts. In this case, the increase of proliferation by mechanotransduction could not happen [61,70]. This hypothesis will be further discussed in the next paragraph.

### 4.2. Perfusion Bioreactor Modelling Based on Experimental Data

According to our experimental results, flow perfusion through the stacked porous hydrogel scaffolds did not trigger the proliferation of the seeded osteoblasts, but the viability of the cells remained high. Therefore, we endeavored to model the physical and chemical environment of the cells within the bioreactors using CFD approaches.

#### 4.2.1. The Issues of Imaging the Hierarchically Porous Bioreactor

In tissue engineering, constructs must be cultured at the centimeter scale with controlled multi-scale architecture. The perfusion bioreactor used during the dynamic cell culture is a multi-scale system showing: (i) spheroids enclosed in (ii) the hydrogel pores, (iii) the macropores formed by the stack of (iv) the hydrogel scaffolds within (v) the bioreactor. 

In order to perform CFD simulations, a 3D image of the bioreactor inside is needed. The first method consists of the serial sectioning of the cellularized scaffold stack and its subsequent histological analysis [6]. However, this method is destructive and time-consuming. The second method is based on X-ray tomography. It was applied on fibroblast tissues cultured on polyacetal bead stacks [39]. The cellular tissues were stained with an osmium tetroxide solution and dried for imaging. The 3D acquisition of the stack was used for LBM simulations [39]. Liu et al. [71] acquired the geometry of porous hydrogel scaffolds (prepared from poly(lactic-co-glycolic) acid and hydroxyapatite, mean pore size of about 325 μm) by X-ray tomography with a resolution of 5.4 μm. They performed CFD simulations of scaffold perfusion. However, X-ray tomography is not suitable for our hydrogel scaffolds, neither in the swollen state nor in the dry state. Indeed, the density difference between the swollen hydrogel and water phases is too low to identify the porous structure from X-ray images. Furthermore, X-ray acquisitions of the dried scaffolds are not relevant since our hydrogel’s structure strongly evolves between the dry state and the swollen state (the pore size decreases from 230 μm to 80 μm [31]). In this study, the perfusion bioreactor was imaged using a T_2_-turbo-RARE sequence (Figure 9B). The cells were labeled with SPION^LA-HSA^ to increase the MRI contrast of the cells [27]. This labeling was persistent within the perfusion bioreactor after 18 days of culture under dynamic conditions (Figure 9B). The grayscale images (with 55 μm resolution) provided by MRI were processed and segmented. Then, it was possible to quantify the total cell volume and deduce the number of cells per hydrogel scaffold using the calibration curve (Figure 7D). However, this estimate is significantly higher (seven times) than the value assessed by confocal microscopy and reconstituted from stacks of images. MRI also gave the geometry of the scaffold stack used as input in the CFD simulations (Figure 10). The pores of the hydrogel scaffolds were not captured by this method because of compromise between spatial resolution and acquisition time (see Appendix A). The permeability of the porous hydrogel was estimated at $K_{hydrogel}≲10^{-6}$ mm^2^ from Kozeny-Carman law [72] with porosity and wetted perimeter evaluated from CLSM images. The permeability of the scaffold stack was estimated at $K_{stack}≅5\times 10^{-3}$ mm^2^. The flow is called “direct perfusion” within the stack and “indirect perfusion” within the microporous scaffolds [73]. Because $K_{hydrogel}$ was three orders of magnitude lower than $K_{stack}$, we deduced that the velocity of the fluid inside the porous hydrogel was much lower than that of the fluid in the macropores of the scaffold stack. Therefore, neglecting the fluid flow inside the porous hydrogel seems reasonable. It is worth noting that this assumption is consistent with the experimental results that indicated a lack of osteoblastic cell proliferation, which calls into question the direct stimulation of cells by the fluid shear stress [19].

#### 4.2.2. Hydrodynamics and Dissolved Oxygen Transport Simulations

In the present work, the average macroporosity of the scaffold stack in the bioreactor, as assessed by MRI, was 21% and varied between 11% and 30%. This value is twice lower as the macroporosity reported by Thibeaux et al. [39] for another type of scaffold, i.e., stacks of 2 mm diameter beads, and also lower than the macroporosity after three weeks of dynamic cell culture reported by Chabanon et al. [6]. The wall shear stress and the fluid velocities were expected to be very heterogeneous [74]. A representative portion of the bioreactor (6.55 mm high) with a porosity of 24 ± 2% was extracted to perform CFD simulations using LB methods (Figure 10). LBMs are suitable for CFD within complex geometries such as perfusion bioreactors [39,75]. The maximal fluid velocity was assessed at 27.29 mm s^−1^, 19 times higher than the perfusion velocity $v_{s}$. The average wall shear stress experienced by the scaffolds is similar to that reported by Thibeaux et al. [39] for the same perfusion velocity but with higher porosity, i.e., 25 mPa and 20 mPa, respectively. Such values were reported to increase cell proliferation [39,76], while a wall shear stress higher than 600 mPa was detrimental for the cells, according to Leclerc et al. [77]. Our results indicated that the hydrogel scaffolds in the bioreactor were not exposed to wall shear stress higher than 100 mPa. However, since the spheroids were confined in the pores of hydrogel scaffolds, they did not experience the same wall shear stress as the external surface of the scaffolds. Consequently, the wall shear stress exerted on spheroids is expected to be several orders of magnitude lower than the typical values computed by LBM (according to the permeabilities estimated in Section 4.1).

The oxygen diffusion coefficient in the hydrogel was an essential input to further simulate the transport of dissolved oxygen in the bioreactor. A dedicated setup was designed to measure it, resulting in an oxygen diffusion coefficient value of $D=(1.6\pm 0.5){\times 10}^{-9}$ m^2^ s^−1^. This value is 47% lower than the diffusion coefficient of oxygen in pure water and is close to that reported by Hulst et al. [34] for alginate beads. Moreover, it falls in the same range as most biological tissues [29]. The partition coefficient of oxygen between the hydrogel and the water phase was also identified. We found a value of K=9±1. This value is unexpectedly high since the water content of our hydrogel is of the order of 95% (*w*/*w*) when swollen in DPBS [31]. Such high values of the partition coefficient have been reported for 2-hydroxyethyl methacrylate (HEMA)-based hydrogels (developed for soft contact lenses) [78]. However, in the latter case, the hydration was significantly lower, i.e., about 40% (*w*/*w*). Further investigations (both experimental and theoretical) are needed to confirm and explain the present behavior. For the numerical simulations of oxygen transport, we assumed K=1.

Our experimental configuration significantly differs from that typically encountered in bone or cartilage tissue engineering bioreactors, where the porous scaffolds are made of impermeable materials. This latter case was addressed by Sacco et al. [21], Hossain et al. [22], and Mehrian et al. [23], among others. Oxygen concentration maps were computed for various perfused scaffolds such as polyetherurethane foams, arrays of circular strands, and open porous titanium scaffolds. In contrast to those matrices, our experimental setup, i.e., spheroids embedded in hydrogel scaffolds and cultured in a bioreactor, is similar to that commonly used for the culture of hepatocyte and tumor spheroids. Several studies dealt with the numerical simulation of oxygen transport in hepatic spheroids encapsulated in hydrogel scaffolds [79,80,81,82]. In particular, Bhise et al. [81] and Sharifi et al. [82] investigated numerically the case of encapsulated hepatic spheroids cultured in a perfusion bioreactor (liver spheroid-on-a-chip model). Generally, the previous studies indicated that the diffusion step may strongly limit oxygen consumption by the cells. In the bioreactor, the transport of dissolved oxygen from the culture medium to the cells is a three-step process, i.e., (i) convective transport in the macroporosity of the bioreactor, (ii) diffusional transport in the hydrogel scaffolds, then (iii) diffusional transport and simultaneous consumption within the spheroids.

The Biot number denoted Bi, allows comparing the diffusion mass transfer resistance in the hydrogel to the convective mass transfer resistance in the perfusion liquid. The convective mass transfer coefficient of dissolved oxygen ($k_{m}$) can be evaluated for our bioreactor from a correlation established by Seguin et al. [83] for a similar packed bed (see Appendix A). We found $k_{m}=3.5\times 10^{-5}$ m.s^−1^. As the mean spacing between spheroids within a scaffold was of the order of l=0.4 mm, we estimate $\mathrm{Bi}=k_{m}l/D=15$ (Bi is even greater when the scaffold half-thickness or its radius is taken as characteristic length) and deduce that the diffusion in the hydrogel is the mass transfer-controlling step. Since Bi≫1 and the oxygen consumption rate by the spheroids represents at the most 5% of the oxygen mass flow rate entering the bioreactor (see Appendix A), it appears reasonable to assume that the oxygen concentration of the liquid phase within the macropores is approximately homogeneous and equal to $c_{0}$, the oxygen concentration at the bioreactor inlet. Our oxygen transport model relies on this assumption. We could thus limit our simulations to the transport of oxygen in the hydrogel and its consumption in the spheroids, avoiding the CPU-intensive simulation of the oxygen transport in the liquid phase. Indeed, since the Peclet number characterizing the convective oxygen transport, i.e., $Pe=1.5\times 10^{5}$, is much larger than the Reynolds number of the liquid flow in the macropores, i.e., Re=15, the computation of oxygen transport in the liquid phase would have required even smaller lattice spacing and time step. Last, the fact that Bi≫1 and only 5% of total oxygen entering the bioreactor is consumed by the cells makes our parametric study focusing on a scaffold highly relevant.

Our simulations at the bioreactor scale point out that the oxygen concentration stays above $K_{M}$ (the Michaelis–Menten constant associated with oxygen consumption law) in 98% of the total volume of the scaffolds. This result is consistent with the cell viability results when cultured under dynamic conditions (Figure 6). Nevertheless, Figure 10 maps (B1,B2) exhibit substantial variations of dissolved oxygen concentration within the hydrogel scaffolds, i.e., from 0.21 mol m^−3^ at the liquid/hydrogel interface down to 0.01 mol m^−3^ deep in a scaffold. This result correlates with the value of the second Damkhöler number that compares the oxygen consumption rate by the cells of a spheroid to the diffusive mass transfer rate of oxygen in the hydrogel, i.e., $\mathrm{Da}=\left(V_{max}d_{s}/\vartheta \right)/\left(Dc_{0}/l\right)=4.5$ (Da is higher when the scaffold is half-thickness or its radius is taken as characteristic length). Low concentration gradients within hydrogel would require Da≪1. When Da≫1, the diffusion step may strongly limit the oxygen consumption by the cells. This condition was met in some of our simulations at the scaffold scale (Figure 11): large spheroids (400 μm in diameter) and/or high cell concentration (900,000 cells per scaffold) led to hypoxic spheroids. 

Taken together, our findings suggest that the number of seeded cells per scaffold, the spheroid size, and the scaffold size and arrangement in the bioreactor are the parameters that can be tuned to optimize the availability of oxygen and the viability of cells within perfused bioreactors. The hydrogel used in our study has the remarkable property of promoting the spontaneous formation of spheroids inside its pores within a few hours after seeding. As also demonstrated before, by adjusting the pore size within the hydrogel scaffold [84], the size of the spheroids can be controlled independently of the number of seeded cells. The numerical tools developed to describe the flow and the oxygen consumption by cells enabled us to study the influence of multiple parameters, such as the perfusion flow rate, the nature of the biomaterial, the number of cells, and the spheroids' diameter on cell viability and oxygen availability. Using the predictive nature of numerical simulations in terms of mass transfer of species such as oxygen, this paper thus provides guidelines for the choice of initial cell culture conditions to ensure optimal and homogeneous cell viability within the whole spheroid-containing hydrogel. The predictive nature of digital simulations also makes it possible to determine the perfusion flow rate of the bioreactor to promote mass transfers in an optimal manner.

Collectively, these findings enabled us to define the optimal conditions inside the bioreactor for an efficient in vitro cell organization and survival in spheroids, which are paramount to future applications with organoids.

## 5. Conclusions

This experimental and numerical study demonstrated the ability of the perfusion bioreactor to maintain the viability of osteoblast spheroids within a stack of porous hydrogel scaffolds. The dynamic culture conditions in the bioreactor provided cells with an oxygen-rich environment. The used method of measuring the oxygen diffusion coefficient in the hydrogel can be extended to other biomaterials applied for 3D cell culture. Cell labeling with SPION^LA-HSA^—combined with MRI analysis, allowed us to rebuild a 3D numerical twin of the seeded bioreactors. The developed numerical tools and parametric analyses enabled us to define the optimal conditions inside the bioreactor for efficient in vitro cell organization and survival in spheroids. These findings advance the field of bone tissue engineering, making it possible in the future to consider producing autologous osteogenic grafts from hydrogel biomaterials seeded with osteoblast or stem cells and cultured in a perfusion bioreactor. The production system could furthermore serve as an in vitro experimental model for drug screening or safety testing of new drugs. The use of the in vitro model developed here could thus help reduce animal testing. In order to validate the proposed strategy, in our future work, we will focus on the quantification of proteins that are early or late markers of osteoblast differentiation, such as ALP, BSP, osteocalcin, osteopontin, collagen type 1 in the perfused 3D cultures. We also aim to study the expression of mRNA expression for Runx2, BSP, OCN, ALP, and COL I genes in dependence on the flow conditions. These detailed characterizations will be a pre-requisite to developing applications for in vitro drug screening with organoids and, in the future, for in vivo bone repair requiring robust and reproducible conditions.

## Figures and Tables

**Figure 1 bioengineering-10-00849-f001:**
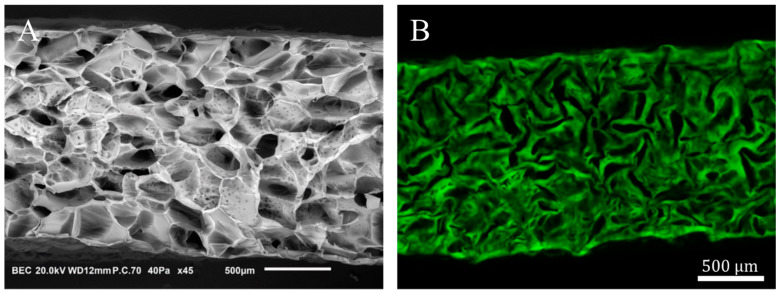
Porous polysaccharide-based hydrogel scaffold. (**A**): XZ cross-section of a freeze-dried scaffold acquired by scanning electron microscopy. (**B**): CLSM XZ cross-section of a porous hydrogel scaffold swollen in DPBS. The cross-sections (**A**,**B**) are not whole in the X-direction.

**Figure 2 bioengineering-10-00849-f002:**
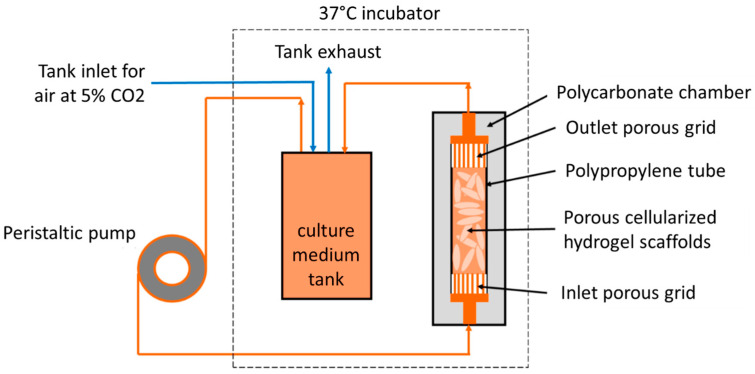
Schematic diagram of the perfusion bioreactor system.

**Figure 3 bioengineering-10-00849-f003:**
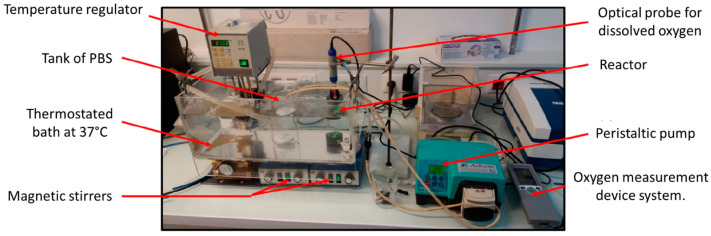
Picture of the experimental device for measuring the diffusion coefficient of oxygen in the hydrogel.

**Figure 4 bioengineering-10-00849-f004:**
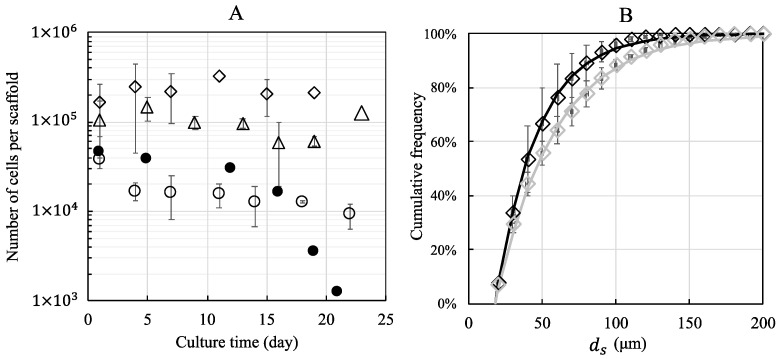
The number of MC3T3E1 preosteoblasts and cumulative distribution of spheroid diameters within porous hydrogel scaffolds under static (full symbols) and dynamic conditions (empty symbols). The initial cell seeding was 100,000 (empty and full circles), 300,000 (empty triangles), and 600,000 (empty diamonds) cells per scaffold. (**A**): The number of cells per scaffold as a function of culture time, evaluated from 370-μm depth measurement and extrapolated to the total thickness of the scaffold (1.4 mm). (**B**): The cumulative (number-average) distribution of spheroid diameters for an initial seeding of 600.000 cells, 24 h post-seeding (black diamonds) and after 14 days of dynamic culture (grey diamonds). Data points were fitted by Equation (1) using least squares method. The error bars represent the standard deviation of the mean calculated for 3 scaffolds from the same bioreactor (*n* = 3).

**Figure 5 bioengineering-10-00849-f005:**
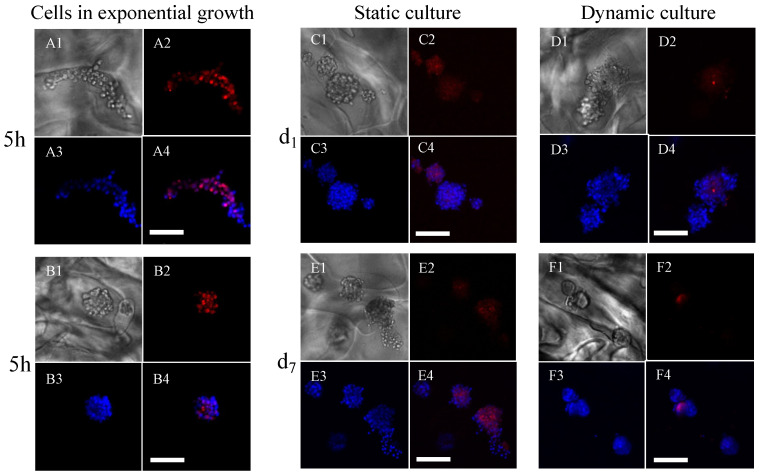
Expression of Ki67 in MC3T3E1 preosteoblasts cultured within porous hydrogel scaffolds under static or dynamic culture conditions. (**A1**–**A4**): Individual cells in the exponential growth phase, fixed with 4% PFA before seeding ((**A1**): white light picture, (**A2**): Ki67 expression, (**A3**): cell nuclei, and (**A4**): merging of images (**A2**,**A3**)). (**B1**–**B4**): Spheroids, formed spontaneously 5 h post-seeding (the image numbering, i.e., 1–4, has the same meaning as above). (**C1**–**C4**,**D1**–**D4**): The scaffolds were removed after 1 day of culture (d_1_). (**E1**–**E4**,**F1**–**F4**): The scaffolds were removed after 7 days (d_7_). (**C1**–**C4**,**E1**–**E4**): static cell culture. (**D1**–**D4**,**F1**–**F4**): dynamic cell culture. Cell nuclei were labeled DAPI (blue). The pictures were acquired by CLSM. The scale bars correspond to 100 μm.

**Figure 6 bioengineering-10-00849-f006:**
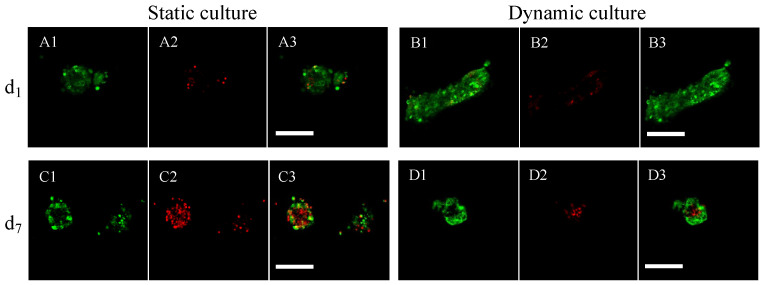
Assessment of MC3T3E1 preosteoblast viability within porous hydrogel scaffolds under static or dynamic cell culture conditions. (**A1**–**A3**): Viability was assessed after 1 day (d_1_) of static cell culture condition ((**A1**): green channel corresponding to the viability signal, (**A2**): red channel corresponding to the cell death signal, and (**A3**): merging of the two channels). (**B1**–**B3**): Viability assessed after 1 day (d_1_) of dynamic cell culture condition (the image numbering, i.e., 1–3, has the same meaning as above). (**C1**–**C3**): Viability assessed after 7 days (d_7_) of static cell culture condition. (**D1**–**D3**): Viability assessed after 7 days (d_7_) of dynamic cell culture condition. (**A1**–**D3**) are magnifications of representative spheroids with a 1.25 μm resolution. The white scale bars correspond to 100 μm.

**Figure 7 bioengineering-10-00849-f007:**
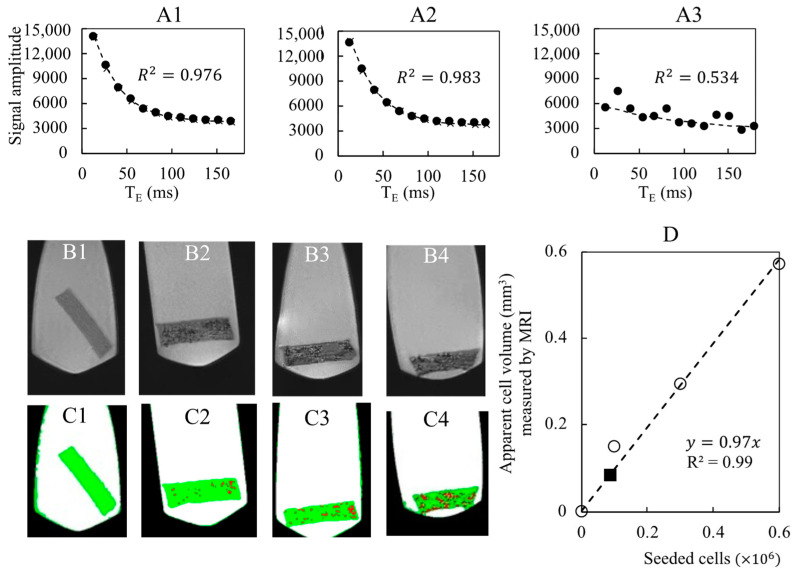
Detection of spheroids and cell number quantification by MRI within porous hydrogel scaffolds. The porous hydrogel scaffolds were seeded with 0 (**B1**,**C1**), 100,000 (**B2**,**C2**), 300,000 (**A1**,**B3**,**C3**) and 600,000 (**A2**,**B4**,**C4**) cells that were pre-labelled with SPION^LA-HSA^ (20 μg_[Fe]_ cm^−2^) and fixed with PFA 4% solution. (**A1**–**A3**): Cell number was assessed by relaxometry using T_2_^*^-MSME sequences (250-μm thick slices with 55-μm resolution; 11 echo times (T_E_) with 13,845 ms spacing). The relaxometry curves were obtained after averaging the signal over the whole scaffolds (**A1**,**A2**) or for a target voxel (**A3**). Data points were fitted by Appendix A according to Milford et al. [33]. R^2^ stands for the linear correlation coefficient. The seeded scaffolds were analyzed by MRI (T2-turbo-RARE sequence, 250-μm thick slice with 55-μm resolution). The grayscale images (**B1**–**B4**) were segmented by image processing into four phases: air (black), liquid (white), hydrogel (green), and cells (red) (**C1**–**C4**). A calibration curve ((**D**), dashed line, and white circle) is obtained by plotting the apparent cellular volume measured by MRI as a function of the number of seeded cells. The black square refers to the bioreactor with stacked hydrogels.

**Figure 8 bioengineering-10-00849-f008:**
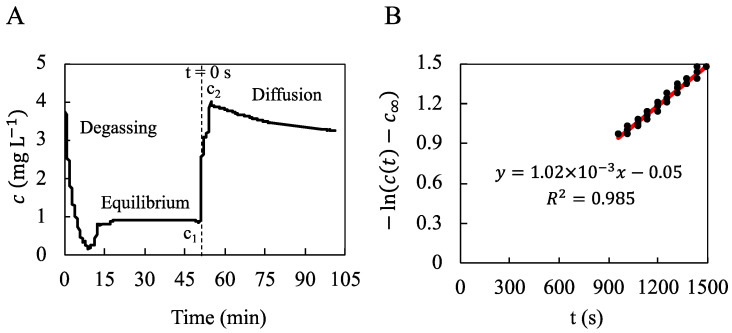
Measurement of the oxygen concentration in a well-stirred reactor to determine the oxygen diffusion coefficient within the hydrogel. (**A**): The oxygen concentration c (mg L^−1^) of the liquid phase in the well-stirred reactor was measured by an oxygen sensor as a function of time. N_2_-bubbling for 10 min allowed for degassing of the oxygen dissolved in the liquid phase (DPBS) until c=0.18 mg L^−1^ (“Degassing”). The well-stirred reactor was closed for 40 min until oxygen concentration stabilized at $c_{1}=0.90$ mg L^−1^ (“Equilibrium”). The liquid phase was quickly renewed with oxygen-saturated DPBS until the maximal oxygen concentration $c_{2}=4.02$ mg L^−1^. Then, the dissolved oxygen diffused from the liquid phase to the hydrogel for 50 min until the final oxygen concentration $c_{\infty }=3.25$ mg L^−1^ (“Diffusion”). (**B**): The diffusion coefficient D was deduced from the linear regression of the quantity $-\ln\left(c(t)-c_{\infty }\right)$ against the time t between t=840 s and t=1500 s. The origin of time t is indicated in (**A**) and corresponds to the beginning of the stepwise concentration variation from $c_{1}$ to $c_{2}$.

**Figure 9 bioengineering-10-00849-f009:**
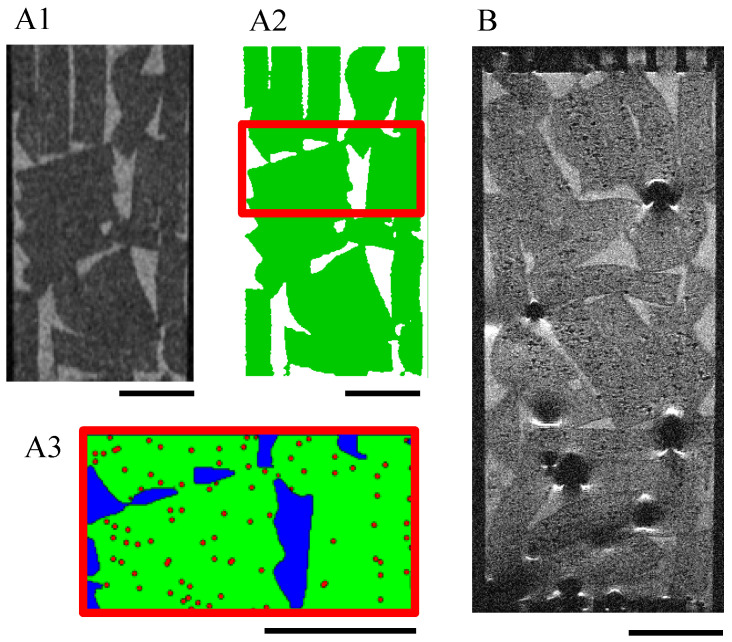
MRI acquisitions of a scaffold stack within a perfusion bioreactor. T_2_-turbo-RARE sequences were performed on slices of 250-μm thickness and 55-μm resolution. (**A1**–**A3**): Bioreactor without cells: the grayscale images (**A1**) were segmented into the fluid (white) and the hydrogel scaffolds (green) (**A2**). The volume of interest (6.55 mm high) defined for the LBM simulations is delimited by red lines. The hydrogel scaffolds (green) of this volume were numerically seeded with spheroids (red dots) of 135 μm diameter (**A3**). The black scale bars stand for 4 mm. (**B**): Initial seeding was 100,000 cells per scaffold; the cells were pre-labeled with SPION^LA-HSA^, 20 μg_[Fe]_ cm^−2^. Cells were fixed with PFA 4% after culturing for 18 days in dynamic conditions (10 mL min^−1^). Cell volume estimation is reported on the calibration curve in Figure 7D. The scale bar stands for 4 mm.

**Figure 10 bioengineering-10-00849-f010:**
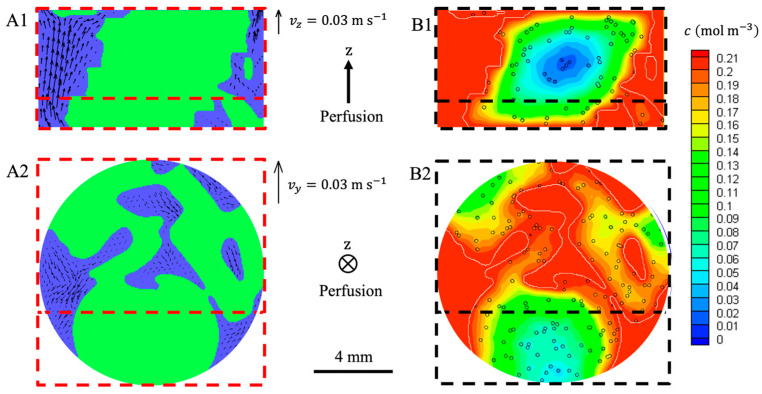
LBM simulations of hydrodynamics and oxygen transport within a 6.55 mm high section of the perfusion bioreactor. The scaffold was numerically seeded with spheroids of 135 μm diameter (400,000 cells per scaffold). The fluid superficial velocity was 1.47 mm s^−1^ (q=10 mL min^−1^). The fluid (blue) had access to the porosity between the stacked hydrogel scaffolds (green). Fluid velocity was assumed to be zero in the hydrogel. Oxygen concentration in the fluid was set to $c_{0}=0.21$ mol m^−3^. The oxygen diffusion coefficient inside the hydrogel scaffolds was $D=1.6\times 10^{-9}$ m^2^ s^−1^. Oxygen consumption by the cells was modeled by a Michaelis-Menten-like kinetics (Equation (8)). The xz cross-section (**A1**,**B1**) and the xy cross-section (**A2**,**B2**) intercept along the dashed red line. (**A1**): The vectors stand for the ($v_{x}$,$v_{z}$) components of the fluid velocities. (**A2**): The vectors stand for the ($v_{x}$,$v_{y}$) components of the fluid velocities. (**B1**,**B2**): The dissolved oxygen concentration in the fluid and the hydrogel is colored according to the linear heat scale.

**Figure 11 bioengineering-10-00849-f011:**
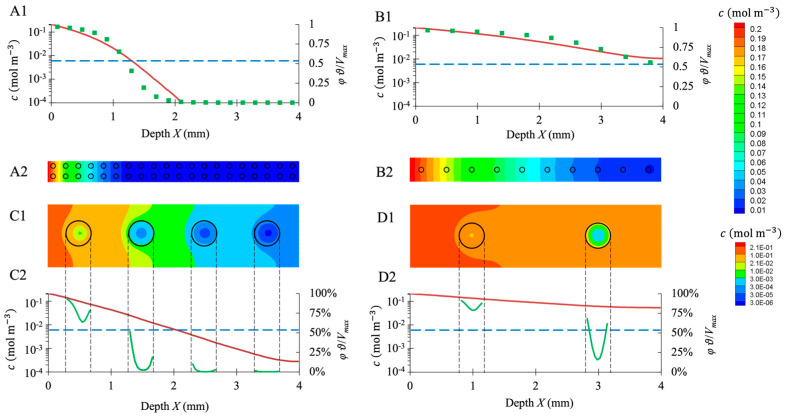
LBM simulations of oxygen transport inside seeded hydrogel scaffolds. Four cases (**A**–**D**) are reported with two different seedings, i.e., 900,000 cells per scaffold (**A1**,**A2**,**C1**,**C2**) and 100,000 cells per scaffold (**B1**,**B2**,**D1**,**D2**), and two spheroid diameters, i.e., 80 μm (**A1**,**A2**,**B1**,**B2**) and 400 μm (**C1**,**C2**,**D1**,**D2**). (**A1**–**D1**): the oxygen concentration profile along the symmetry axis between two spheroid rows is plotted in red lines; the blue lines correspond to $c=K_{M}/2$ and $=0.5\;V_{max}/\vartheta$, i.e., the hypoxia limit. The green squares (**A1**,**B1**) represent the oxygen consumption rate at the spheroid center (expressed as a percentage of $V_{max}/\vartheta$), the green lines (**C1**,**D1**) the oxygen consumption rate along the spheroid diameter. (**A2**–**D2**): oxygen concentration map in the scaffold median plane displayed in linear heat scale (**A2**,**B2**) or exponential heat scale (**C2**,**D2**). The black circles correspond to the spheroid outline.

## Data Availability

Not applicable.

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
