# Peer review of "Perfusion of MC3T3E1 Preosteoblast Spheroids within Polysaccharide-Based Hydrogel Scaffolds: An Experimental and Numerical Study at the Bioreactor Scale"

_bioengineering, 2023, doi:10.3390/bioengineering10070849_

Round 1
Reviewer 1 Report
Manuscript
Title: “Perfusion of MC3T3E1 preosteoblast spheroids within polysaccharide-based hydrogel scaffolds: experimental and numerical study at the bioreactor scale"
Author(s): Jérôme Grenier, Bertrand David, Clément Journé, Iwona Cicha, Didier Letourneur and Hervé Duval
Reviewer Comments to Author(s)
Recommendation: Accept
The issues have been addressed and the manuscript can be accepted in the present form
Author Response
Reply in the attached file.

Reviewer 2 Report
This manuscript deals with porous polysaccharide-based hydrogel scaffolds developed in a bioreactor-type perfusion device that generates favorable mechanical stresses while enhancing nutrient transfers.
The authors combined advanced experiments and theory in a unique way and the whole manuscript is impressive. However, this work could be further improved.
POINTS FOR IMPROVEMENT:
1. The abstract could be shorten summarizing the findings of this work.
2. Write a paragraph reporting summarizing applications of this work and directions of future work.
3. Please, add few schemes illustrating a) the followed experimental methodology of this work b) the experimental set up.
Author Response
Reply in the attached file.

Reviewer 3 Report
Review (recommended major revision)
This modelling study explores a novel investigation approach to an in vitro tissue engineering process by integrating the external stimuli sensors into 3D culture reactor systems. By using hydrogel template scaffolds, a bioreactor (device) and integration, favorable mechanical loading was achieved.
***recommended major revision***

Review (recommended major revision)
This modelling study explores a novel investigation approach to an in vitro tissue engineering process by integrating the external stimuli sensors into 3D culture reactor systems. By using hydrogel template scaffolds, a bioreactor (device) and integration, favorable mechanical loading was achieved.
***recommended major revision***
Author Response
See the reply in the attached file.
